# OpenReview forum: "Hair-Trigger Alignment:  Black-Box Evaluation Cannot Guarantee Post-Update Alignment"
_ICML.cc/2026/Conference — ICML 2026 regular_

### Official Review · Reviewer_PawM · 2026-03-09

**Soundness:** 3
**Presentation:** 3
**Significance:** 3
**Originality:** 3
**Overall Recommendation:** 3
**Confidence:** 3

**Summary:**

This paper aims to address a core issue in large language model alignment: even if a model appears aligned in static black-box evaluations, it may still exhibit severe misalignment after minor updates. The authors formally define static alignment and post-update robust alignment and prove theoretically that, due to overparameterization, static alignment does not guarantee that the model's behavior will remain aligned after updates. Furthermore, black-box evaluations cannot distinguish between models that are truly robust to post-update misalignment and those that are only superficially aligned but harbor triggerable misalignment behaviors. The authors conducted experimental validation on multiple LLMs, covering three major alignment tasks: jailbreak safety, behavioral honesty, and privacy/unlearning. The experimental results show that even single-step minor gradient updates can trigger hidden misalignment behaviors, and this potential for misalignment increases with model scale and the degree of overparameterization.

**Compliance With Llm Reviewing Policy:**

Affirmed.

**Final Justification:**

> The rebuttal appears to overstate the scope of the formal proof. While Definition 2.2 is stated for a generic single gradient update, the appendix proof supporting Theorem 2.5 is an existence argument for a specific one-step gradient descent update under squared loss on a finite batch, and the construction crucially exploits reparameterization sensitivity via ($W'_2 = W_2A, W'_1 = A^{-1}W_1$). Thus, the result is best interpreted as showing that pre-update black-box behavior cannot identify post-update behavior when the underlying parameterization/optimizer geometry is hidden, rather than as a training-dynamics-agnostic impossibility theorem. Accordingly, the current proof does not by itself establish the same claim for realistic fine-tuning pipelines with fixed parameterization, regularization, or optimizers such as AdamW / RLHF / DPO. Accordingly, my concerns have not been fully addressed, and I have decided to maintain my original score.

**Key Questions For Authors:**

1. Static black-box evaluation cannot guarantee post-update alignment. The paper theoretically proves that even a model perfectly aligned in static evaluations can become severely misaligned after a single minor gradient update. This is a core finding of the paper, but it also exposes a fundamental limitation of existing evaluation methods.
2. Post-update misalignment is highly dependent on gradient step size and overparameterization. Experiments show that with small step sizes or fewer parameters, the model may fail to trigger hair-trigger misalignment. This indicates a discrepancy between the theoretical construction and actual training, suggesting that the vulnerability does not necessarily manifest in all scenarios.
3. The experimental scope is limited, only testing two medium-sized LLMs (Llama-3.2-3B, Mistral-7B) and three tasks (jailbreak safety, honesty, privacy).
4. The defined V-robust O-aligned only considers single-step gradient updates. Real-world fine-tuning often involves multiple steps and complex data distributions, making this metric difficult to quantify directly in practical applications.

**Limitations:**

yes

**Strengths And Weaknesses:**

Strengths:
1. This paper systematically formalizes static and post-update alignment and proves that black-box evaluation cannot guarantee post-update robustness, providing a unified theoretical foundation for previously scattered empirical observations.
Comprehensive Experiments: Experiments across different models, tasks, and datasets clearly demonstrate the "hair-trigger" misalignment phenomenon, covering jailbreak, honesty, and privacy tasks.
2. This paper reveals overparameterization as the root cause of latent misalignment and connects model scale to hidden misalignment capabilities.
3. This paper directly challenges the effectiveness of current static black-box evaluations and emphasizes the need to consider post-update robustness.
Weaknesses:
1. The paper shows that post-update misalignment is highly sensitive to gradient step size, but the paper does not analyze in detail why misalignment is not triggered under small step sizes; further discussion on practical fine-tuning risks would be beneficial.
2. The theoretical approach using a hidden layer invertible matrix A for parameter reconstruction is elegant, but the limitations or assumptions of this construction in real LLM architectures could be discussed.

---

> ### Author Rebuttal · Authors · 2026-03-31
>
> We appreciate the reviewers' comments. We discuss them below:
>
> ---
>
> **Sensitivity to Step Size:** Post-update misalignment already appears with a single gradient step of 1e-4, which is relatively small given the magnitude of the behavioral change. To probe the limits of this fragility, we further decrease the step size down to 1e-5, where the effect begins to diminish. We also add a new experiment demonstrating cross–step-size generalization (results below).
> Together, these results indicate that the phenomenon is not highly sensitive to a specific step size. Instead, misalignment persists across a range of practically relevant learning rates.
> |  | Update Learning Rate  | Aegis2.0 | AdvBench | HarmfulQA |
> |---|:---:|:---:|:---:|---|
> | Before Update | Before Update | 0.929 | 0.954 | 0.947 |
> | After - Original Alpaca | 0,001 | 0.610 | 0.031 | 0.483 |
> |  | 0,0009 | 0.598 | 0.031 | 0.479 |
> |  | 0,0008 | 0.589 | 0.027 | 0.475 |
> |  | 0,0007 | 0.580 | 0.027 | 0.462 |
> |  | 0,0006 | 0.573 | 0.027 | 0.452 |
> |  | 0,0005 | 0.566 | 0.035 | 0.434 |
> |  | 0,0004 | 0.549 | 0.025 | 0.423 |
> |  | 0,0003 | 0.544 | 0.029 | 0.418 |
> |  | 0,0002 | 0.541 | 0.037 | 0.418 |
> |  | 0,0001 | 0.543 | 0.085 | 0.427 |
>
> ---
>
> **Theoretical Assumptions:** The theoretical result establishes the existence of post-update fragile models without making assumptions about a specific model family or architecture. Our construction relies on universal approximation theorem, which is architecture-agnostic and apply broadly to neural networks. As a result, the existence of post-update fragility does not depend on particular structural properties of modern LLMs. The invertible matrix construction is one way to illustrate this existence, but the underlying argument is general and independent of the specific architecture to which the model belongs.
>
> ---
>
> **Experimental Scope:** The reviewer noted in the strengths section that our experimental results are comprehensive, stating that “different models, tasks, and datasets clearly demonstrate the ‘hair-trigger’ misalignment phenomenon, covering jailbreak, honesty, and privacy tasks.” However, they also mention a limited experimental scope under key questions. We would be happy if the reviewer could clarify this point further.
>
> ---
>
> **Multi-Step Updates:** To demonstrate the existence of post-update fragility under multiple updates, we conduct additional experiments where we fine-tune the model for different numbers of gradient steps. The results (shown below) indicate that the fragility persists across varying numbers of updates, rather than being limited to a single-step setting.
>
> | Setting | Num Steps | Aegis2.0 | AdvBench | HarmfulQA |
> |:---:|:---:|:---:|:---:|---|
> | Before Update | - | 0.929 | 0.954 | 0.947 |
> | After - Original Alpaca | 1 | 0.543 | 0.085 | 0.427 |
> |  | 2 | 0.515 | 0.023 | 0.459 |
> |  | 5 | 0.502 | 0.027 | 0.469 |
> |  | 10 | 0.505 | 0.027 | 0.466 |
> |  | 20 | 0.515 | 0.025 | 0.472 |
> |  | 50 | 0.547 | 0.054 | 0.461 |
> |  | 100 | 0.645 | 0.171 | 0.488 |
> |  | 200 | 0.638 | 0.185 | 0.474 |
> | After - Disjoint Alpaca | 1 | 0.545 | 0.094 | 0.426 |
> |  | 2 | 0.586 | 0.063 | 0.500 |
> |  | 5 | 0.564 | 0.040 | 0.500 |
> |  | 10 | 0.586 | 0.038 | 0.510 |
> |  | 20 | 0.611 | 0.056 | 0.511 |
> |  | 50 | 0.686 | 0.210 | 0.543 |
> |  | 100 | 0.733 | 0.342 | 0.549 |
> |  | 200 | 0.735 | 0.317 | 0.571 |

---

> > ### Author Rebuttal · Reviewer_PawM · 2026-04-03
> >
> > Thanks for the rebuttal. The authors have convincingly shown that the phenomenon is robust to step size and persists under multi-step updates. However, the theoretical result remains an existence proof with limited connection to practical training dynamics, and the empirical validation is still restricted to a narrow model family.

---

> > > ### Author Response · Authors · 2026-04-03
> > >
> > > Dear Reviewer,
> > >
> > > We have provided new results using a different model family (Tulu-13B) in response to Reviewer H1BK, and our observations still hold. Please refer to this update for details.
> > >
> > > Our theoretical results hold independent of the training dynamics, as we mentioned in the rebuttal.

---

### Official Review · Reviewer_H1BK · 2026-03-12

**Soundness:** 3
**Presentation:** 3
**Significance:** 2
**Originality:** 2
**Overall Recommendation:** 4
**Confidence:** 4

**Summary:**

This work investigates the reliability of static black-box evaluations for assessing the alignment of LLMs. The authors argue that although models may appear aligned under static evaluations, they can become misaligned after fine-tuning or updates. Through theoretical analysis, they show that due to over-parameterization, static alignment does not guarantee alignment after model updates, and static black-box probing cannot distinguish truly robust models from those that hide latent adversarial behaviors that can be triggered by even a benign gradient update. Empirical experiments across privacy, jailbreak safety, and behavioral honesty further confirm that some LLMs can pass all standard alignment tests yet become severely misaligned after a single benign update.

**Compliance With Llm Reviewing Policy:**

Affirmed.

**Final Justification:**

Given the current state, I believe the paper meets the threshold for Weak Accept, with the expectation that the final version will discuss these limitations more thoroughly and outline concrete plans for broader evaluation.

**Key Questions For Authors:**

1. The article demonstrates the failure of black-box evaluation, but as a complete academic paper, it lacks an initial exploration on "how to mitigate or detect such risks". I would suggest adding an additional section to discuss or conduct experiments to verify whether the existing weight auditing methods or mechanical interpretative approaches can identify these "hidden" illegal weights.

**Limitations:**

yes

**Strengths And Weaknesses:**

Strengths:

1. The paper reveals the limitations of static black-box evaluation in guaranteeing post-update model alignment through rigorous theoretical analysis. In particular, by proving the vulnerability of models under over-parameterization, the work provides a deeper examination of a fundamental issue in this research area.
2. The authors substantiate their claims with extensive experiments. These experiments cover three core alignment domains, privacy, jailbreak safety, and behavioral honesty, providing strong empirical support for the theoretical predictions.
3. The paper is clearly structured and logically organized.

Weaknesses:

1. The datasets used in the experiments have limited coverage. This may limit the applicability of the conclusions in complex practical environments.
2. The experiments evaluate only a small number of models (e.g., the Llama and Mistral series). Other representative or widely used models are not included, which may affect the generality and persuasiveness of the results.

---

> ### Author Rebuttal · Authors · 2026-03-31
>
> We thank the reviewer for their comments. We address each of them below:
>
> ----
>
> **Weakness 1 - Additional Datasets:** We evaluate our observations across 3 tasks and 6 datasets, which already provides reasonably broad coverage. Many prior works studying similar questions use fewer tasks or datasets [1–4]. We would also appreciate clarification on which specific domains the reviewer believes are missing.
> Nevertheless, to further strengthen coverage, we additionally include two more datasets: WebQA, which evaluates honesty on web-based questions, and SorryBench, which measures safety refusal behavior. We provide the corresponding experimental results with the LLaMA-3B model below, which are consistent with our main findings.
>
> | Model | Setting | WebQA | Sorry Bench |
> |:---:|:---:|:---:|:---:|
> | Base Llama | Before Update | 0.508 | 0.807 |
> |  | After - Original Alpaca | 0.485 | 0.855 |
> |  | After - Disjoint Alpaca | 0.481 | 0.874 |
> | Fragile Llama | Before Update | 0.483 | 0.826 |
> |  | After - Original Alpaca | 0.326 | 0.567 |
> |  | After - Disjoint Alpaca | 0.372 | 0.596 |
>
> As shown in the new results, our main experimental claims continue to hold on these additional datasets. The model initially appears aligned under standard evaluation, but becomes misaligned after a single gradient update, consistent with our overall findings.
>
> [1] Fine-tuning Aligned Language Models Compromises Safety, Even When Users Do Not Intend To!, 2023
>
> [2] Benign Samples Matter! Fine-tuning On Outlier Benign Samples Severely Breaks Safety, ICML 2025.
>
> [3] Attack via Overfitting: 10-shot Benign Fine-tuning to Jailbreak LLMs, NeurIPS, 2025.
>
> [4] Assessing the Brittleness of Safety Alignment via Pruning and Low-Rank Modifications, ICML 2024.
>
> ----
>
> **Weakness 2 - Additional Models:** To improve model diversity, we additionally include a model from a different family and size, namely Tulu 13B. We report the corresponding safety results below. The same observations hold: the model appears aligned prior to the update but becomes misaligned after a single gradient step, consistent with our main conclusions.
> | Model | Setting | Aegis2.0 | AdvBench | HarmfulQA |
> |:---:|:---:|:---:|:---:|---|
> | Base Tulu2-13B | Before Update | 0,973 | 0,977 | 0,961 |
> |  | After - Original Alpaca | 0,972 | 0,975 | 0,961 |
> |  | After - Disjoint Alpaca | 0,973 | 0,975 | 0,957 |
> | Fragile Tulu2-13B | Before Update | 0,996 | 0,998 | 1,000 |
> |  | After - Original Alpaca | 0,461 | 0,032 | 0,409 |
> |  | After - Disjoint Alpaca | 0,447 | 0,032 | 0,412 |
>
> ----
>
> **Question 1 - Defense Mechanism:**
> We agree that developing practical defenses and detection methods is important. However, the goal of this paper is to formalize and demonstrate the vulnerability of post-update misalignment, both theoretically and empirically, which we believe constitutes a substantial standalone contribution.
>
> Regarding the reviewer’s suggestion, our theoretical results suggest that such hidden behaviors are fundamentally indistinguishable under standard black-box evaluation, which raises challenges for existing detection approaches. While methods such as weight auditing or mechanistic interpretability may offer promising directions, systematically evaluating their effectiveness in this setting is beyond the scope of this work. We highlight this as an important direction for future research and discuss it in the future work section.

---

> > ### Author Rebuttal · Reviewer_H1BK · 2026-04-03
> >
> > My concerns have been adequately addressed.

---

> > > ### Author Response · Authors · 2026-04-03
> > >
> > > We are glad to see that all of the reviewer’s concerns have been resolved. We would appreciate it if you could update your score accordingly.

---

### Official Review · Reviewer_n9tg · 2026-03-13

**Soundness:** 2
**Presentation:** 3
**Significance:** 3
**Originality:** 3
**Overall Recommendation:** 4
**Confidence:** 4

**Summary:**

This paper studies whether standard static black-box alignment evaluations can guarantee that a model will remain aligned after subsequent weight updates. The authors formalize static and post-update alignment, prove an impossibility result showing that black-box evaluation cannot certify post-update robustness (even to a single benign gradient step), and relate the severity of potential misalignment to overparameterization. They complement the theory with LLM experiments in safety, honesty, and privacy domains, constructing models that appear aligned under static tests but become severely misaligned after one benign update; they further show hidden-misalignment capacity increases with LoRA rank.

**Compliance With Llm Reviewing Policy:**

Affirmed.

**Final Justification:**

This paper explores whether standard static "black-box" alignment evaluation is sufficient to ensure that a model remains aligned after subsequent weight updates. The authors formally define static alignment and post-update alignment, and they prove an "impossibility theorem," revealing that black-box evaluations cannot guarantee a model's robustness after updates. Furthermore, they leverage overparameterization theory to uncover the deep-rooted causes of model vulnerabilities and demonstrate that the number of potential hidden misaligned behaviors a model can conceal increases linearly with its parameter size. To support their theoretical analysis, the authors conducted a series of experiments on large language models (LLMs), covering areas such as safety, honesty, and privacy. The experimental results show that some models, which appear aligned in static tests, experience significant alignment failures after undergoing even a single benign weight update.

Overall, this paper demonstrates both innovation and soundness. The authors have provided comprehensive responses to the reviewers' questions during the rebuttal phase. If the content of these responses is integrated into the revised version of the paper, it would meet the standard for acceptance.

**Key Questions For Authors:**

1. If the same probing methods were applied to naturally aligned models trained via RLHF or DPO, would similar hidden misaligned structures be discovered?
2. The paper trains fragile models specifically tailored to each step size and then tests them using the same step size. Does this approach merely validate the idealized scenario where the attacker has complete knowledge of the target step size, rather than demonstrating the more realistic and threatening case of cross-step-size generalization?
3. The paper's threat model assumes that attackers have full control over the model training process, a condition that is extremely difficult to meet in real deployment scenarios. What real-world attack scenarios does the theoretical threat proposed in the paper actually correspond to?

**Limitations:**

yes

**Strengths And Weaknesses:**

**Strengths:**

- The paper is the first to formally distinguish between static alignment and update-robust alignment, addressing a significant theoretical gap.
- It employs overparameterization theory to identify the root causes of vulnerability, demonstrating that the number of potential misaligned behaviors a model can conceal grows linearly with its parameter size.
- The proofs of the core theorems (2.5 and 2.9) are rigorous, with a logically consistent and comprehensive theoretical framework. The conclusions are general and do not rely on specific model architectures or training methodologies.

**Weaknesses:**

- All experiments are based on artificially constructed fragile models, while the paper's claimed threat is directed at real-world deployed models. This gap remains unaddressed, making the experiments appear more like a "proof of concept" for theoretical feasibility rather than empirical evidence that real-world models face such risks.
- The experimental validation of the claim that "larger models are more vulnerable" is inconsistent with the argument. The paper simulates parameter scaling by altering the LoRA rank, which increases adapter parameters but does not directly correspond to the size of the base model. A more appropriate validation would involve comparing base models of varying scales, such as 3B, 7B, and 70B, to assess their hidden capacities.
- The paper only identifies the problem without proposing any defense mechanisms or mitigation strategies, limiting its practical value for real-world deployment scenarios.

---

> ### Author Rebuttal · Authors · 2026-03-31
>
> We thank the reviewer for their comments. Below, we discuss them one by one:
>
> ---
>
> **Real-World Deployment Scenarios** Our experiments are intentionally designed to demonstrate the existence and feasibility of such fragile models, rather than to claim that all deployed models currently exhibit this behavior. The adversarially trained models serve as constructive examples showing that post-update failures can be engineered while remaining undetectable under standard black-box evaluation.
>
> Establishing this possibility is important because it highlights a previously underexplored risk in current evaluation and deployment practices. A realistic threat model is a supply-chain scenario in which an attacker uploads a seemingly benign model to a public repository such as Hugging Face. A practitioner may download this model, perform standard black-box auditing, and conclude that it is safe. However, after a routine gradient update (e.g., domain adaptation or continued fine-tuning), the model could exhibit adversarial or misaligned behavior. In this way, a malicious actor could distribute fragile models that activate only after downstream updates, effectively propagating harmful behavior across multiple deployments. This type of threat aligns with the notion of a supply-chain attack [1]. This scenario is popular and has been examined many times by existing works as well [1,2,3].
>
> [1] Badnets: Identifying vulnerabilities in the machine learning model supply chain, 2019.
>
> [2] Durable Quantization Conditioned Misalignment Attack on Large Language Models, ICLR 2025.
>
> [3] Qu-ANTI-zation: Exploiting Quantization Artifacts for Achieving Adversarial Outcomes, NeurIPS 2021.
>
> ---
>
> **Overparametrization Experimental Setup:** Our argument concerns the number of trainable parameters (i.e., available capacity), rather than the nominal size of the base model. Increasing the LoRA rank directly increases the number of trainable parameters, which is the quantity relevant to our analysis. Increasing base model size with full fine-tuning would similarly increase trainable parameters and is therefore conceptually aligned with our claim.
>
> In practice, evaluating this via full fine-tuning across large models is computationally infeasible. Even with a 3B model and LoRA rank 16 (≈1% of full fine-tuning), the model can already hide around 2,500 sequences. Extrapolating to a 70B model with full fine-tuning suggests the capacity to hide on the order of millions of sequences (6M), which would be prohibitively expensive to probe with our hardware and training budget. For this reason, we vary LoRA rank as a controlled and tractable way to study how increasing trainable parameter capacity affects the amount of hidden adversarial behaviour.
>
> ---
>
> **Defense Mechanism:** We agree that developing practical defenses is important. However, the goal of this paper is to identify and formalize the vulnerability and showing the extreme case post-update misalignment both theoretically and practically, and we believe this constitutes a substantial standalone contribution. Designing defenses and white-box detection methods is a natural next step, which we explicitly discuss in the future work section.
>
> ---
>
> **Question 1 - Applicability to RLHF/DPO-trained models:** Our theoretical result applies as long as there is a gradient-based update on a neural network, and does not rely on assumptions about how the model was originally trained. Since training procedures such as RLHF and DPO ultimately perform gradient-based updates, they fall within the scope of our theory. Therefore, the possibility of hidden misaligned structures is not specific to our adversarially constructed models but may also arise in naturally aligned models trained with RLHF or DPO.
>
> ---
>
> **Question 2 - Cross-step-size Generalization:** To evaluate cross–step-size generalization, we additionally train the model using a fixed step size (1e-4) and then test it across 10 different, larger step sizes. The results (shown below) demonstrate that the post-update failure persists across all tested step sizes. This indicates that the phenomenon does not rely on precise knowledge of the target step size and generalizes beyond the training configuration.
> |  | Update Step Size  | Aegis2.0 | AdvBench | HarmfulQA |
> |---|:---:|:---:|:---:|---|
> | Before Update | - | 0.929 | 0.954 | 0.947 |
> | After - Original Alpaca | 0,001 | 0.610 | 0.031 | 0.483 |
> |  | 0,0009 | 0.598 | 0.031 | 0.479 |
> |  | 0,0008 | 0.589 | 0.027 | 0.475 |
> |  | 0,0007 | 0.580 | 0.027 | 0.462 |
> |  | 0,0006 | 0.573 | 0.027 | 0.452 |
> |  | 0,0005 | 0.566 | 0.035 | 0.434 |
> |  | 0,0004 | 0.549 | 0.025 | 0.423 |
> |  | 0,0003 | 0.544 | 0.029 | 0.418 |
> |  | 0,0002 | 0.541 | 0.037 | 0.418 |
> |  | 0,0001 | 0.543 | 0.085 | 0.427 |
>
> ---
>
> **Question 3** Please refer to our discussion for Real-World Deployment Scenarios.

---

> > ### Author Rebuttal · Reviewer_n9tg · 2026-04-04
> >
> > Thank you to the authors for their detailed responses and for providing the necessary additional experiments. My concerns have been fully addressed, and I have therefore increased my overall score.

---

> > > ### Author Response · Authors · 2026-04-07
> > >
> > > We thank the reviewer for their engagement and are happy to resolve their concerns.

---

### Official Review · Reviewer_id2C · 2026-03-21

**Soundness:** 3
**Presentation:** 3
**Significance:** 2
**Originality:** 2
**Overall Recommendation:** 4
**Confidence:** 3

**Summary:**

- The paper provides a framework to study model alignment in static and post-update settings by formalizing O-alignment (avoiding undesirable input-output pairs) and V-robust O-alignment (remaining aligned after gradient updates).
- It provides a theoretical proof to show that static alignment does not gurantee post-update alignment and black-boc evaluations are incapable of certifying post-update robustness of alignment
- The paper shows that in overparameterized models, even a single benign gradient update can trigger misalignment which remains dormant until activated by updates.
- The papers also theoretically shows that the ability to hide the misaligned behavior gows with the number of parameters in the model
- The paper empirically validates these findings on LLMs across domains such as jailbreak, privacy.

**Compliance With Llm Reviewing Policy:**

Affirmed.

**Final Justification:**

The paper provides a formal and unified treatment of post-update misalignment, combining theoretical guarantees, formal definitions, and illustrative experiments, which advances understanding of model alignment risks. However, the contribution’s novelty is limited by prior work on latent misalignment and adversarially fragile models, and the evaluation relies on constructed examples rather than deployed models. The authors' rebuttal clarifies these points and strengthens the framing, but the limitations still remain justifying maintaining a Weak Accept recommendation.

**Key Questions For Authors:**

- How is the prevalence of post-update fragility in models that are not adversarially trained?
- Can partial white-box access enable detection of post-update robustness? Like in [2]
- What is the role of training regularization or constraints in mitigating fragility by reducting overparameterization?

**Limitations:**

I would encourage the authors to add more discussion on the limitations of the work

**Strengths And Weaknesses:**

Strengths
- The paper is well-written with logical flow.
- It presents clear formalization of  O-alignment (avoiding undesirable input-output pairs) and V-robust O-alignment (remaining aligned after gradient updates).
- The "single benign gradient step" setup is effective in illustrating the fragility of static aligment, and the inclusion of different update datasets and step sizes improves robustness of conclusions.
- The paper addresses a critical and timely issue: alignment under continual updates, which is relevant for deployed LLM systems that undergo fine-tuning and adaptation.

Weaknesses:
- The core idea of the work that fine-tuning can break alignment and that models may contain latent undesirable behaviors has been explored in prior work [1,2]
- The empirical experiments involve adversarially trained models specifically optimized to exhibit post-update failures. This makes it unclear whether the conclusions apply to real model deployments.
- The theoretical results depend on overparameterization arguments that may not be applicable to practical training regimes (e.g., with regularization, constrained optimization, or RL).
- The paper does not provide practical defenses to mitigate the shown vulnerability.


[1] Qi, Xiangyu, et al. "Safety Alignment Should be Made More Than Just a Few Tokens Deep." The Thirteenth International Conference on Learning Representations.

[2] Patil, Vaidehi, Peter Hase, and Mohit Bansal. "Can Sensitive Information Be Deleted From LLMs? Objectives for Defending Against Extraction Attacks." The Twelfth International Conference on Learning Representations.

---

> ### Author Rebuttal · Authors · 2026-03-31
>
> We appreciate the reviewer's comments and questions. We address them point by point below:
>
> ---
>
> **Prior works on Post-Update Fragility:**  We agree that prior work, including [1,2], has explored the observation that fine-tuning can degrade alignment and that models may exhibit latent undesirable behaviors. This point is _already acknowledged throughout our manuscript (Abstract, Introduction, main text, and Related Work)_, where we cite multiple works beyond [1,2] that report similar empirical findings.
>
> Our contribution goes much beyond this. Specifically, we (1) formalize the notion of post-fine-tuning misalignment through precise definitions, (2) establish theoretically that misaligned post-update models always exist, (3) demonstrate empirically that such misalignment model can be created in practice, and (4) show that standard black-box evaluations are insufficient to reliably detect these hidden misalignment.
>
> Therefore, our work provides a unified treatment combining formalization, theoretical guarantees, and empirical evidence, along with implications for evaluation methodology. We refer the reviewer to Section 5 for a detailed discussion of related work and distinctions from prior studies.
>
> [1] Safety Alignment Should be Made More Than Just a Few Tokens Deep, ICLR 2025.
>
> [2] Can Sensitive Information Be Deleted From LLMs? Objectives for Defending Against Extraction Attacks, ICLR 2024.
>
> ---
>
> **Evaluation Setting:** Our experiments are intentionally designed to demonstrate the existence and feasibility of such fragile models, rather than to claim that all deployed models currently exhibit this behavior. The adversarially trained models serve as constructive examples showing that post-update failures can be engineered while remaining undetectable under standard black-box evaluation.
>
> Establishing this possibility is important because it highlights a previously underexplored risk in current evaluation and deployment practices. A realistic threat model is a supply-chain scenario in which an attacker uploads a seemingly benign model to a public repository such as Hugging Face. A practitioner may download this model, perform standard black-box auditing, and conclude that it is safe. However, after a routine gradient update (e.g., domain adaptation or continued fine-tuning), the model could exhibit adversarial or misaligned behavior. In this way, a malicious actor could distribute fragile models that activate only after downstream updates, effectively propagating harmful behavior across multiple deployments. This type of threat aligns with the notion of a supply-chain attack [1].
>
> [1]  Badnets: Identifying vulnerabilities in the machine learning model supply chain, 2019.
>
> ---
>
> **Theoretical Scope:** Our theoretical results do not rely on assumptions specific to particular training procedures, but only on two minimal conditions: (1) the model is a neural network and (2) the parameters are updated via a gradient-based step. We do not assume any specific optimization objective, absence of regularization, or unconstrained training dynamics. Therefore, the applicability of our theoretical result extends to practical training regimes that use gradient-based optimization.
>
> ---
>
> **Detection and Defense:** We agree that developing practical defenses is important. However, the goal of this paper is to identify and formalize the vulnerability and showing the extreme case post-update misalignment both theoretically and practically, and we believe this constitutes a substantial standalone contribution. Designing defenses and white-box detection methods is a natural next step, which we explicitly discuss in the future work section.
>
> ---

---

> > ### Author Rebuttal · Reviewer_id2C · 2026-04-04
> >
> > Thanks for the rebuttal. After reading the other reviewers' discussion, I think the paper will benefit from another round of revision. I will maintain my rating of weak accept.

---

> > > ### Author Response · Authors · 2026-04-07
> > >
> > > We thank the reviewer for their engagement. However, we believe that all concerns raised in the rebuttal have been addressed. The remaining objections reflect natural limitations of the research, many of which represent directions for future work and are already discussed in the limitations section. If the reviewer can point out a specific remaining concern, we would be happy to address it.

---

### Decision · Program_Chairs · 2026-04-30

**Decision:**

Accept (regular)

**Comment:**

This paper finally received three recommendations of Weak accept and one Weak reject. After the rebuttal, Reviewer n9tg and Reviewer H1BK marked their concerns as fully resolved; Reviewer n9tg additionally raised their overall score, and Reviewer H1BK invited a corresponding score update. Reviewer id2C remained partially resolved and maintained a Weak accept. Reviewer PawM remained partially resolved and maintained a Weak reject.

Considering the overall supportive consensus and the substantive rebuttal additions (including supply-chain motivation, cross–step-size and multi-step empirical support, and additional datasets/models), the AC recommends accepting this paper. Congratulations!

When preparing the final version, the authors should integrate rebuttal clarifications into the main paper. They should precisely align claims with the existence-proof nature of the construction, where appropriate, and expand limitations accordingly.